# Internal Defense System of *Mytilus galloprovincialis* (Lamarck, 1819): Ecological Role of Hemocytes as Biomarkers for Thiacloprid and Benzo[a]Pyrene Pollution

**DOI:** 10.3390/toxics11090731

**Published:** 2023-08-25

**Authors:** Alessio Alesci, Davide Di Paola, Angelo Fumia, Sebastian Marino, Claudio D’Iglio, Sergio Famulari, Marco Albano, Nunziacarla Spanò, Eugenia Rita Lauriano

**Affiliations:** 1Department of Chemical, Biological, Pharmaceutical, and Environmental Sciences, University of Messina, 98166 Messina, Italy; aalesci@unime.it (A.A.); dipaolad@unime.it (D.D.P.); marino.sebastian1999@gmail.com (S.M.); cladiglio@unime.it (C.D.); serfamulari@unime.it (S.F.);; 2Department of Clinical and Experimental Medicine, University of Messina, Padiglione C, A. O. U. Policlinico “G. Martino”, 98124 Messina, Italy; angelofumia@gmail.com; 3Department of Veterinary Sciences, University of Messina, 98168 Messina, Italy; malbano@unime.it

**Keywords:** hemocytes, pollution, biomarker, internal defense system

## Abstract

The introduction of pollutants, such as thiacloprid and benzo[a]pyrene (B[a]P), into the waters of urbanized coastal and estuarine areas through fossil fuel spills, domestic and industrial waste discharges, atmospheric inputs, and continental runoff poses a major threat to the fauna and flora of the aquatic environment and can have a significant impact on the internal defense system of invertebrates such as mussels. Using monoclonal and polyclonal anti-Toll-like receptor 2 (TLR2) and anti-inducible nitric oxide synthetase (iNOS) antibodies for the first time, this work aims to examine hemocytes in the mantle and gills of *M. galloprovincialis* as biomarkers of thiacloprid and B[a]P pollution and analyze their potential synergistic effect. To pursue this objective, samples were exposed to the pollutants, both individually and simultaneously. Subsequently, oxidative stress biomarkers were evaluated by enzymatic analysis, while tissue changes and the number of hemocytes in the different contaminated groups were assessed via histomorphological and immunohistochemical analyses. Our findings revealed that in comparison to a single exposure, the two pollutants together significantly elevated oxidative stress. Moreover, our data may potentially enhance knowledge on how TLR2 and iNOS work as part of the internal defense system of bivalves. This would help in creating new technologies and strategies, such as biosensors, that are more suitable for managing water pollution, and garnering new details on the condition of the marine ecosystem.

## 1. Introduction

Industrialization and new techniques of intensive agriculture have resulted in a steady and exponential increase in the production of waste, which is often discharged or accumulated in water systems eventually flowing into rivers and seas [1]. Water pollution represents an alteration of its original characteristics through the introduction of anthropogenic contaminants, i.e., various chemical and toxic substances, such as biocides, pesticides, cosmetics, and pharmaceuticals, altering its use for human nutrition and/or sustenance of biotic communities [2]. Moreover, many of these substances, slowly degrading, remain for long periods and can also bioaccumulate, thus enhancing their dangerousness in the aquatic environment [3,4,5,6].

Thiacloprid [(Z)-3-(6-chloro-3-pyridylmethyl)-1,3-thiazolidin-2-ylidencianamide], a type of neonicotinoid insecticide, is highly selective in action and has excellent systemic activity against sucking and biting insects such as whiteflies and aphids [7]. As a potent agonist of insect nicotinic acetylcholine receptors, thiacloprid can bind to gamma-aminobutyric acid (GABA) receptors on the postsynaptic membrane. In addition, it can disrupt signal transduction in the insect central nervous system [8,9]. Neonicotinoids are currently the most widely used and sold insecticides in the world and provide effective pest control [10,11]. The massive use of these pesticides first contaminates the soil of the treated crops before transferring residues to the aquatic environment. Moreover, recent monitoring studies in numerous nations have shown that these pollutants have contaminated streams, rivers, and lakes all across the world, with residue levels in the low g/L (ppb) range [10,12].

Polycyclic aromatic hydrocarbons (PAHs) are highly detected environmental pollutants and widely distributed in the marine environment, introduced into the aquatic systems of urbanized coastal and estuarine areas through fossil fuel spills, domestic and industrial waste discharges, atmospheric inputs, and continental runoff [13].

Benzo[a]pyrene (B[a]P) is an isomer belonging to the (PAHs) and is characterized by a pyrene condensed with a phenyl group (C_20_H_12_). Among the members of its class, it is certainly the most studied compound for its hepatotoxic, carcinogenic, mutagenic, teratogenic, and immunosuppressive effects, which can affect wild and farmed aquatic animals through the trophic chain [14]. Aquatic organisms are widely used as models for the study of contaminants in marine and freshwater ecosystems because they can assimilate pollutants in numerous ways, including ingestion or filtration of particles suspended in water, ion exchange of dissolved heavy metals through lipophilic membranes (gills), and surface adsorption tissues [15,16]. Histopathological changes are used as biomarkers to assess the overall health status of aquatic animals exposed to pollutants [2]. Bivalve mollusks, particularly mussels, are considered good bioindicators of water pollution. First, this is due to their wide distribution, sedentary life mode, and high abundance, and second, because of their filter-feeding mode, ability to accumulate high levels of substances in their organs and high tolerance to abiotic changes [17]. Mussels also attract attention concerning the assessment of human health risks associated with water deterioration [18]. Several studies highlighted a vast amount of toxic effects described in aquatic animal models following exposure to benzo[a]pyrene (B[a]P), including oxidative stress, cytotoxicity, mutagenicity, tumor formation, and immune repression [19,20]. In addition, B[a]P is highly genotoxic to both marine and freshwater species, following the IARC hazard range, being classified in Group 1 for its extreme health risk [21], although its genotoxic susceptibility has been studied mainly in freshwater species [22]. Several studies have investigated the presence of B[a]P in aquatic organisms as biomarkers of toxicity [23,24]. In addition, B[a]P has also been found in bivalves (*Mytilus galloprovincialis*, Lamarck 1819), cephalopods (*Todarodes sagittatus*, Lamarck 1798), crustaceans (*Nephrops norvegicus*, Linnaeus 1758), and fish (*Mullus barbatus* Linnaeus 1758, *Scomber scombrus* Linnaeus 1758, *Micromesistius poutassou* Risso 1827, *Merluccius merluccius* Linnaeus 1758) in several basins from the central Adriatic Sea [25].

The internal defense system of mussels, as of all invertebrates, lacks the highly specific adaptive immunity of jawed vertebrates characterized by lymphocytes and the production of immunoglobulin antibodies. However, they protect themselves from microbes, parasites, and environmental threats by circulating blood cells responsible for the production of reactive oxygen and nitrogen species and phagocytosis, i.e., highly conserved processes [26]. These defense mechanisms include pattern recognition receptors (PRRs) such as Toll-like receptors (TLRs), NOD-like receptors (NLRs) [27], and phagocytosis [28]. Phagocytic cells observed in invertebrates bear different names (amoebocytes, coelomocytes, or hemocytes) depending on the host species, but exhibit morphology attributable to vertebrate macrophages with relatively comparable functions [29]. Several studies have analyzed defense cells as biomarkers in fish [30,31,32] and in invertebrates [33].

This study aims to evaluate hemocytes in the mantle and gills of *M. galloprovincialis* as biomarkers of thiacloprid and B[a]P pollution and investigate their possible synergistic effect using mono- and polyclonal anti-TLR2 and anti-iNOS antibodies for the first time.

## 2. Materials and Methods

### 2.1. Contaminants

Thiacloprid (purity 99.9%) was purchased as a powder from Sigma-Aldrich (Prague, Czech Republic). Distilled water was used as a solvent to prepare a stock solution (1 mg L^−1^), and from this, dilutions were made to obtain the concentration of 4.5 μg/L^−1^. B[a]P was purchased from Sigma-Aldrich (Saint Louis, MO, USA) and was put into solution according to the manufacturer’s instructions and was brought to a concentration of 1 μg/L^−1^.

### 2.2. Selection of Concentrations

This study aims to assess the potential synergic effect of the combination of two of the contaminants most found in the aquatic environment at concentrations likely to overlap with those already detected. Based on previous studies [34], we selected a concentration of 4.5 μg/L^−1^ thiacloprid as it is similar to that found in the environment, and similarly, we selected a 1 μg/L concentration of B[a]P as reported from previous studies [35,36].

### 2.3. Animal Preparation

In the present study, *M. galloprovincialis* was used as a model organism by which to assess the combined effect of a chemical mixture of thiacloprid and B[a]P. Mussels were purchased from a local farm of bivalves at Faro Lake (Company Faro SRL, Frutti di Mare, Messina, Italy). A total of 320 mussels (mean length = 6.21 ± 0.14 cm) were housed in the laboratory for acclimation for 10 days before conducting the experiments.

### 2.4. Experimental Design

Healthy mussels were randomly split into four 40 L experimental tanks. The groups were divided as follows: one control group (n = 40) with only fresh seawater; one group (n = 40) exposed to thiacloprid 4.5 μg/L^−1^; one group (n = 40) exposed to B[a]P 1 μg/L^−1^; and one group (n = 40) exposed to both thiacloprid 4.5 μg/L^−1^ and B[a]P 1 μg/L^−1^. The organisms were maintained under controlled conditions (aeration, temperature 23.03 ± 0.6 °C, pH 7.54 ± 0.16, salinity 31.79 ± 1.37, 1100 m Osm kg^−1^, 12 h light:12 h dark) for 7 days, and the experiment was repeated two times. This assessment of toxicity was performed using a semi-static system with natural seawater. Seawater and stock solutions were refreshed every 48 h to guarantee a constant supply of fresh brackish food and prevent famine while exposed for an extended period. Twenty mussels from each experimental group were randomly collected on the 7th day of exposure. No mortality was found in the control group and the group exposed to thiacloprid 4.5 μg/L^−1^. Both thiacloprid 4.5 μg/L^−1^ and the B[a]P 1 μg/L single concentration exposure group showed no statistically relevant mortality (<3% and 4%, respectively); on the contrary, thiacloprid 4.5 μg/L^−1^ and B[a]P 1 μg/L^−1^ displayed a mortality rate of 8%.

### 2.5. Tissue Preparation

Following the procedures for sample preparation for light microscopy, *M. galloprovincialis* samples were processed. After fixation in Immunofix, the samples were washed in crescent alcohol (up to 100% alcohol) to remove water from the tissues and allow better penetration of paraffin. The samples were washed with xylol, which is solvent for paraffin, to clarify the tissues. Finally, they were included, via inclusion unit, in bioplast [37]. On each slide, 5 μm thick sections were placed that had been cut using a rotary microtome (LEICA 2065 Supercut, Nussloch, Germany). Following deparaffinization in xylene, slides were rehydrated in alcohol solutions ranging from 100% to 30% alcohol, followed by distilled water.

### 2.6. Histological Analysis

The sections were stained using Mallory Trichrome (04–020802, BioOptica Milano S.p.A., Milan, Italy), Masson Trichrome (04–010802, BioOptica Milano S.p.A., Milan, Italy), and Alcian blue/Periodic Acid Schiff (AB/PAS) (04–163802 BioOptica Milano S.p.A., Milan, Italy) [38,39].

### 2.7. Confocal Microscopy

Slices received treatment in a 2.5% solution of bovine serum albumin (BSA) after deparaffinization and rehydration. After that, primary anti-TLR2 and anti-iNOS antibodies were incubated in the sections. The incubation of secondary antibodies was carried out after one night of exposition to the primary antibodies. To prevent photobleaching, the slices were mounted using Fluoromount [40,41]. As a negative control, experiments were carried out without the primary antibodies. Rat gut tissues were used as a positive control to ensure the immunopositivity of the primary antibodies (data not shown). The slices were examined using a confocal laser scanning microscope with a META module (Zeiss LSM DUO, Carl Zeiss MicroImaging GmbH, Jena, Germany). The fluorescence was detected using argon (458 nm) and helium–neon (543 nm) lasers. The images were captured and enhanced using Zen 2011 (LSM 700 Zeiss software, Oberkochen, Germany). To prevent photodegradation, each picture was snapped as rapidly as possible. The digital pictures were added to a figure composite using Adobe Photoshop CC version 2019 (Adobe Systems, San Jose, CA, USA). The fluorescence intensity curves were then evaluated using Zen 2011 “Display profile” feature. Information about the antibodies is provided in Table 1.

### 2.8. Statistical Analysis

Ten sections and twenty fields per sample were examined to collect data for the quantitative analysis. ImageJ 1.53e was used to evaluate the amount and positivity of the cells. SigmaPlot version 14.0 (Systat Software, San Jose, CA, USA) was used to count the number of hemocytes that were TLR2- and iNOS-positive [42]. One-way ANOVA and the Student’s *t*-test were employed to analyze the normally distributed data. The mean values and standard deviations (SD) of hemocytes detected are presented as follows: ** *p* ≤ 0.01, * *p* ≤ 0.05.

### 2.9. Oxidative Stress Biomarkers

Mantle and gills tissues (n = 10 for each experimental group) were defrosted and weighed separately. Briefly, the samples were homogenized on ice, with 180 μL of ice-cold physiological saline. The supernatant was obtained by centrifuging the homogenate at 4000× *g* for 15 min at 4 °C. According to previous studies, the content of superoxide dismutase (SOD), catalase (CAT), and malondialdehyde (MDA) in the supernatant was measured using commercial kits (Nanjing Jiancheng Bioengineering Institute, Nanjing, China) [43,44]. Lipid peroxidation products (measured as MDA) were investigated via the thiobarbituric acid (TBARS) method, and the MDA concentration was expressed as nanomoles per mg protein as seen in a previous study [45]. One unit of SOD activity was defined as the amount of enzyme required to inhibit the oxidation reaction by 50% and was expressed as U/mg protein. One unit of CAT activity was defined as the amount of enzyme required to consume 1 μmol H_2_O_2_ in 1 s and was expressed as U/mg protein. Protein concentration was assayed using the Bradford assay using bovine serum albumin as a standard.

### 2.10. Data Analysis

All values in the figures and text are expressed as the mean ± standard deviation (SD) of N number of experiments. The results were analyzed via one-way ANOVA followed by a Bonferroni post hoc test for multiple comparisons. The data were tested for normal distribution with Bartlett’s test, and they were represented as mean ± standard deviation (SD), (*p* value of <0.0001). Statistical analysis was performed using Graphpad Prism 9 (Graphpad, Boston, MA, USA).

## 3. Results

### 3.1. Histological Analysis

Histological analysis of the mantle of the control group showed a cuboidal epithelium that becomes columnar in some sections. Most of the mantle edges showed the presence of collagen and muscle. AB/PAS histochemical staining revealed sulfated mucosubstances distributed among the collagen fibers of the mantle margin in the form of agglomerates. Mucosal cells containing acidic (light blue stained) and neutral (magenta purple) mucopolysaccharides were observed in the epithelium. In the treated groups, there is an increase in acidic mucopolysaccharides compared to neutral ones. Mucosal cells with purple secretion, indicating a mixture of acidic and neutral mucopolysaccharides, are evident in the epithelia of pollutant-exposed specimens. The epithelial cells appear less organized, and the connective tissue is looser, especially in the group exposed to both pollutants. There is also a decrease in the presence of cilia in sensory function cells. Infiltrates of hemocytes appear evident in the exposed groups (Figure 1).

The gills of the control specimens show normal morphology, and the presence of well-organized lamellae, with no changes in the epithelia. In contrast, histology of the gills of pollutant-exposed mussels showed a cartilaginous core, lamellar damage, and epithelial changes, sometimes with lamellae fusion. In addition, some epithelial cells were hypertrophic and hyperplastic. Tissue damage appears evident in all exposed groups but is greater in the group exposed to both pollutants, where there is progressive congestion of the lamellae. Moreover, hemocyte infiltrates appear evident in the exposed groups (Figure 2).

### 3.2. Immunohistochemical Analysis

Confocal microscopic investigation revealed hemocytes immunoreactive to TLR2 and iNOS in all groups. Moreover, hemocytes immunoreactive to these molecules, which colocalize conspicuously, are visible in both the epithelium and subepithelium. However, there is a noticeable increase in the number of hemocytes in the exposed groups, especially in the one exposed to both pollutants. Furthermore, markedly positive epithelial cells to both antibodies are noted, especially in the group exposed to B[a]P and thiacloprid. Interestingly, not all hemocytes colocalized for the antibodies tested, suggesting a difference in the function of these cells (Figure 3 and Figure 4).

Statistical analysis revealed an increasing number of immunoreactive hemocytes to TLR2 and iNOS, especially in the group exposed to the synergic effect of thiacloprid and B[a]P (Table 2).

### 3.3. Oxidative Stress Biomarkers and Lipid Peroxidation

SOD, CAT, and MDA activity was not found to be considerably impacted by the thiacloprid 4.5 μg/L^−1^ exposure group in the current investigation (Figure 5 and Figure 6). On the contrary, the B[a]P 1 μg/L^−1^ group presented an increase in SOD, CAT, and MDA activity in both mantle and gills after 7 days of exposition compared to the control group (Figure 5 and Figure 6). Finally, the thiacloprid 4.5 μg/L^−1^ and B[a]P 1 μg/L^−1^ mixture group showed a significant rise in the activity of SOD and CAT in the both gills and mantle compared to the single-exposure groups and the control (Figure 5 and Figure 6).

## 4. Discussion

Environmental pollution, as well as population expansion and resource scarcity, has become a potential risk to humanity. Chemical pollutants, including heavy metals, microplastics, and persistent organic pollutants, enter living organisms through ingestion, inhalation, and contact [46]. Sewage discharge, agriculture, and industrial activities are some of the main sources of contamination in the aquatic environment [47]. As the demand for vegetables, grains, and fruits has increased due to the continued expansion of the world’s population, the use of pesticides to protect crops from disease or predators has also increased [48]. Often, these substances reach freshwater ecosystems, affecting the organisms that inhabit them. Pesticides or insecticides flowing into rivers can affect the morphology, function, and health of organisms [49]. An aquatic ecosystem can be contaminated by a combination of different pesticides and other pollutants, so organisms such as mussels may bioaccumulate more than one substance [50]. Thiacloprid is a neuroactive insecticide belonging to the class of neonicotinoids, which as pesticides similar in structure to nicotine. Neonicotinoids are competitors of acetylcholine, and their mechanism of action is disruption of signal transmission [51,52].

Mussels are sessile bivalve mollusks widely distributed in the world’s oceans and important members of many coastal ecosystems. As filter feeders, they can accumulate and concentrate contaminants from their surroundings, making them effective bioindicators of environmental quality [53]. Hemocytes, the circulating immune cells of mussels, have been proposed as potential biomarkers of environmental stressors due to their sensitivity to changes in the surrounding environment and their role in the immune response. These defense cells are involved in several physiological and immunological processes, including phagocytosis, encapsulation, and production of reactive oxygen species and nitric oxide [54]. These processes can be affected by exposure to pollutants, pathogens, and other environmental stressors. Therefore, changes in hemocyte morphology, function, and gene expression can provide valuable information about the health and condition of mussels and their environment [55].

In this study, we evaluated for the first time the effect of thiacloprid and B[a]P, individually and combined, on hemocytes in the mantle and gills of *M. galloprovincialis* using anti-TLR2 and anti-iNOS antibodies, confirming the ecological role of these cells as environmental biomarkers.

The mantle, gills, and hemocytes of *M. galloprovincialis* play crucial roles in mussel survival, growth, and reproduction. The mantle is a tissue that surrounds the body of the mussel and secretes the shell. It is an important organ for mussel growth, shell formation, and protection from predators. It also plays a role in the uptake and accumulation of pollutants in the body of the mussel [56]. Several studies have shown how environmental contaminants carry a significant increase in peroxide dismutase, catalase, and glutathione peroxidase in the mantle [57,58]. Gills are an important organ for respiration and exchange of gases and nutrients in the body of the mussel. They are also an important site for the uptake and accumulation of pollutants in the mussel’s body. Wang (2018) analyzed the effects of cadmium on the gills of *M. galloprovincialis*, showing that cadmium exposure led to a significant decrease in the activity of antioxidant enzymes in the gills, inducing oxidative stress [59].

Our study showed tissue changes in the mantle and gills as evidenced by histological data. Physiologically, the mantle is composed of an epithelium-rich columnar in cells with secretory activity, with a well-organized connective underneath [60]. The gills normally consist of rows of gill filaments formed by epithelial tissue, aligned to create a homogeneous trabecular structure, typical of gill lamellae, with apical ciliated cells that facilitate water flow, oxygen uptake, and food collection [61]. The epithelia of animals in the pollutant-exposed groups were disorganized and non-homogeneous, with several congested cells and structures. Hemocytes are specialized cells in the body of mussels that play important roles in immune function, wound healing, and nutrient uptake and transport [62]. Stara et al. (2021) analyzed the effects of long-term exposure of *M. galloprovincialis* to thiacloprid, suggesting altered hemocyte and hemolymph biochemical parameters, antioxidants, and ROS production and induced histological alterations in mussel tissues [34]. Hoer (2015) analyzed the effects of exposure to polycyclic aromatic hydrocarbon (PAH) fluoranthene on hemocytes of the blue mussel *Mytilus trossulus* (Gould, 1850). The study found that PAH exposure caused an increase in the number of circulating hemocytes [63]. A study by Karagiannis et al. (2011) analyzed the effects of exposure to the herbicide atrazine on gene expression in hemocytes of the blue mussel *M. galloprovincialis*, finding that exposure to atrazine causes changes in the expression of genes involved in oxidative stress, DNA damage, and apoptosis [64]. Hemocytes also play an ecological role by being involved in the uptake, transport, detoxification, and elimination of pollutants and their metabolites.

B[a]P can be taken up by hemocytes through receptor-mediated endocytosis and can then be transported throughout the bivalve’s circulatory system. This process can lead to the accumulation of B[a]P in various tissues and organs of the bivalve, including the gills and mantle [59]. Frenzilli et al. (2009) analyzed the effects of B[a]P on hemocytes of the clam *Chamelea gallina* (Linnaeus, 1758). The study found that B[a]P exposure resulted in increased hemocyte numbers and production of oxygen and nitrogen free radicals [65]. In addition to B[a]P uptake and transport, hemocytes also play a role in the detoxification and elimination of the pollutant. However, B[a]P metabolites can be more toxic than the parent compound and can induce cell damage and dysfunction in hemocytes [66].

Toll-like receptors (TLRs) are a class of phylogenetically highly conserved transmembrane proteins that play an important role in the innate immune system by recognizing pathogen-associated molecular patterns (PAMPs) and initiating immune responses [67,68,69,70]. We previously demonstrated TLR2 expression in the defense cells of invertebrates [71,72], protochordates [73,74], and several aquatic vertebrates such as myxins [74,75], cartilaginous fish [76], and bony fish [77,78,79]. Inducible nitric oxide synthase (iNOS) is an enzyme that catalyzes the production of nitric oxide (NO) in response to various stimuli, such as pathogens and inflammatory cytokines. iNOS is expressed in a variety of immune cells, including the hemocytes of bivalves such as mussels [80]. In previous studies, we evaluated the expression of iNOS in the defense cells of invertebrates [72] and vertebrates [81,82]. iNOS is also involved in the response to environmental stressors in mussels. A study by Dailianis (2009) analyzed the expression of iNOS in hemocytes of the mussel *M. galloprovincialis* in response to cadmium exposure, finding that iNOS was overexpressed in the hemocytes of mussels exposed to the contaminant [83]. A study by Shi et al. (2012) analyzed the role of iNOS in hemocyte proliferation and metabolism in the scallop *Chlamys farreri* (Müller, 1776), showing that iNOS expression was upregulated in the hemocytes of scallops subjected to immune challenge, and that iNOS inhibition resulted in a significant decrease in hemocyte proliferation and metabolic activity. This suggested that iNOS plays a role in the regulation of cellular metabolism in these organisms [84]. Consistent with these data, we found hemocytes immunoreactive to TLR2 and iNOS in all groups tested. Quantitative analysis showed that the number of hemocytes increased in the group exposed to B[a]P and in the group exposed to thiacloprid. In the group exposed to both pollutants, a significant increase in hemocytes was noted, suggesting a synergistic effect of the two compounds on defense cells. Moreover, interestingly, not all hemocytes colocalized for the antibodies tested, suggesting that hemocytes positive for iNOS might also have a cytotoxic function, while those positive only for TLR2 might perform a mostly phagocytic function, as reported in previous studies [71,72]. Furthermore, TLR2- and iNOS-positive epithelial cells were found in pollutant-exposed groups, suggesting an involvement of these cells in oxidative stress-induced responses. Although the antibodies used are produced in mammals, we could see marked cross-reactivity, also evidenced by the “display profile” function of the confocal microscope. In addition, an interesting response of hemocytes to environmental pollutants emerges from our data on how these cells are able to increase in number and even overexpress recognition molecules implicated in defense systems, thus showing an evolutionary conservation of host defense mechanisms. However, several genomics studies have shown the presence of human- or mammalian-produced immune molecules in invertebrate defense cells [85,86,87,88]. Moreover, cross-reactive immune molecules such as TLRs of human and mammalian origin have been tested and found in several invertebrates [89,90,91].

The use of hemocytes as biomarkers of environmental stressors in mussels has several advantages over traditional biomarkers, such as enzyme activity and tissue analysis. Hemocytes are a non-destructive and minimally invasive sampling method because they can be collected without harming the organism. In addition, hemocytes are highly sensitive to changes in the environment, making them effective indicators of short-term exposure to stressors. However, there are also limitations to the use of hemocytes as biomarkers. Hemocyte responses to stressors can be highly variable, and changes in their morphology and function can be influenced by factors such as age, sex, and reproductive status. Furthermore, the interpretation of hemocyte responses can be complex, as changes in gene expression may reflect both direct effects of stressors and secondary effects of immune activation. Despite these limitations, the use of hemocytes as biomarkers of environmental stressors in mussels is promising. The activity of enzymes involved in the antioxidants system was also herein investigated. Environmental contaminants-induced oxidative stress is thought to be caused by two main mechanisms: an increase in ROS production and a decrease in cellular antioxidant defenses [92,93]. Oxidative stress is primarily brought on by reactive oxygen species (ROS). Since oxygen free radicals can react excessively with SOD and CAT, disrupting the body’s antioxidant defense system, excessive ROS generation in vivo leads to oxidative stress [94]. SOD is an anti-oxidant enzyme that can shield the body from environmental oxidative damage, eliminating ROS and stopping lipid peroxidation. Data collected showed an imbalance in the antioxidant defense system with an evident rise in the levels of SOD and CAT in both the mantle and gills of mussels after 7 days of exposition. Moreover, the levels of antioxidant enzyme were more consistent in the co-exposure group (thiacloprid 4.5 μg/L^−1^ and B[a]P 1 μg/L^−1^) compared to the single exposure groups of thiacloprid and B[a]P.

## 5. Conclusions

Our results showed that the combination of the two contaminants increased oxidative stress strongly compared to the single exposition. Further research is needed in order to standardize sampling protocols and identify the most effective hemocyte-based biomarkers for different types of stressors. Moreover, the use of multi-biomarker approaches, which combine hemocyte-based biomarkers with other traditional biomarkers, may provide a more comprehensive assessment of environmental quality. However, our data may provide useful insights to improve the understanding of the roles of TLR2 and INOS in the internal defense system of bivalves, and their involvement in the management of environmental pollutants. This could enable the development of new strategies and technologies such as the creation of biosensors suitable for improving water pollution management and providing ever-new details on the health of the marine ecosystem.

## Figures and Tables

**Figure 1 toxics-11-00731-f001:**
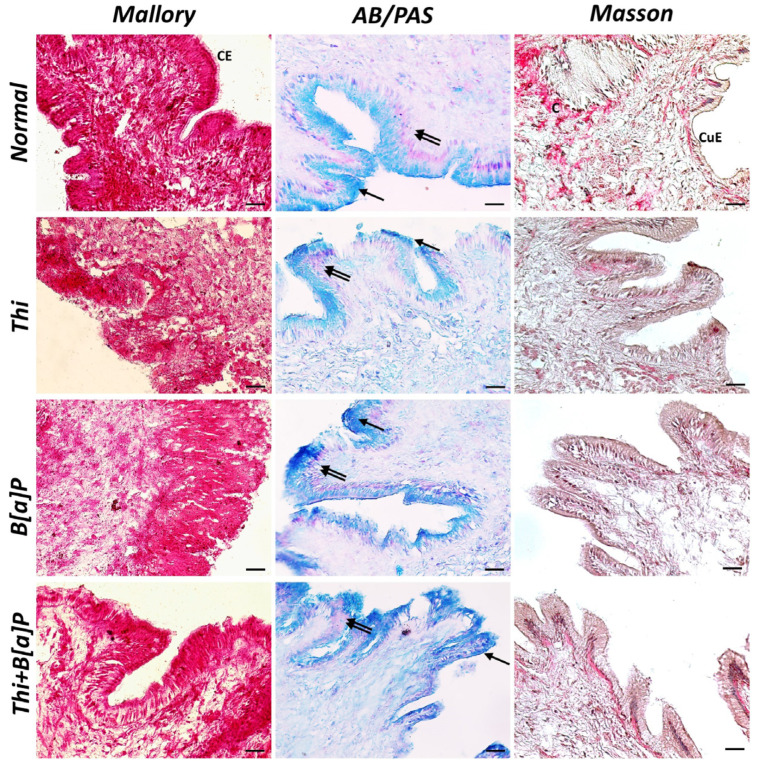
Cross section of mantle of *M. galloprovincialis* (40×, scale bar: 10 µm). Mallory trichrome and Masson trichrome morphological staining revealed a homogeneous, well-organized connective tissue (C) and an epithelium with columnar (CE) and cuboidal (CuE) cells in control animals. AB/PAS histochemical staining revealed the presence of acidic (arrow) and neutral (double arrow) mucopolysaccharides. In groups exposed to pollutants, the epithelium was more disorganized, with looser connective tissue and an overproduction of acidic mucopolysaccharides.

**Figure 2 toxics-11-00731-f002:**
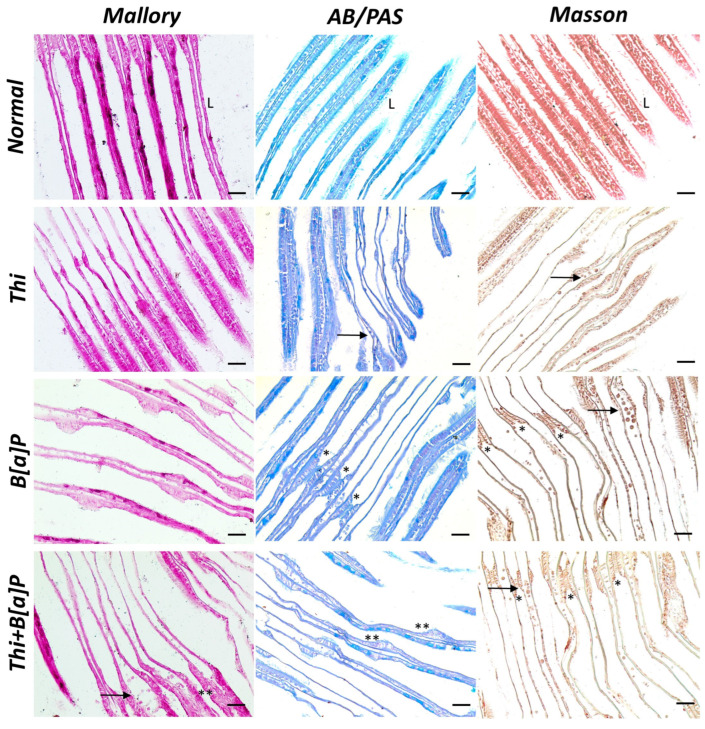
Cross section of gills of *M. galloprovincialis* (40×, scale bar: 10 µm). Control specimens showed well-organized, linear gills with homogeneous lamellae (L) and compact epithelium. In the exposed groups, the lamellae appear disorganized, and the epithelial tissue appear inhomogeneous. Fusion of lamellae (*) with hyperplastic and hypertrophic cells (**) is also evident in some places. In addition, aggregates of hemocytes are evident in the pollutant-exposed groups (arrow).

**Figure 3 toxics-11-00731-f003:**
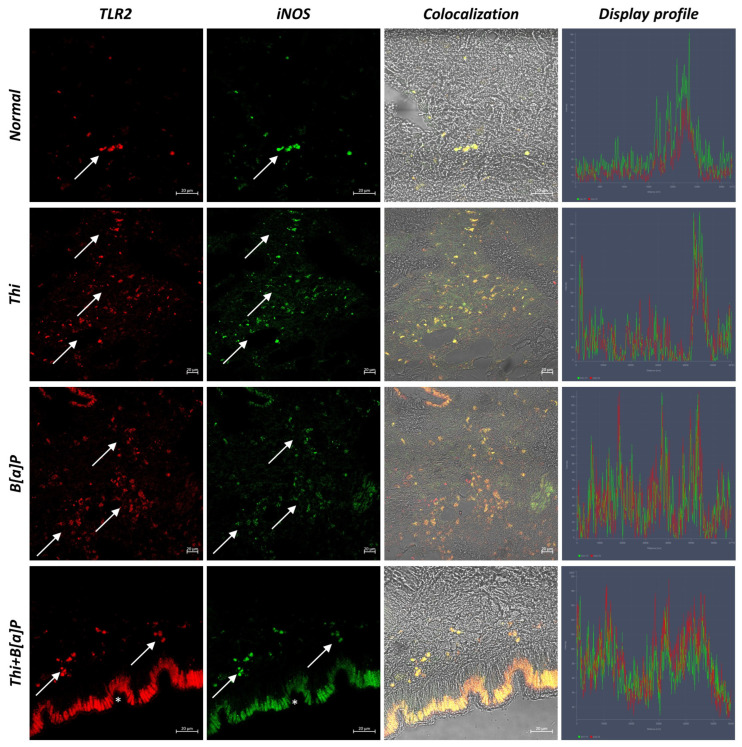
Cross section of mantle of *M. galloprovincialis* (40×, scale bar: 20 µm). Confocal microscopy analysis shows hemocytes positive for TLR2 (red) and iNOS (green) (arrow). Not all hemocytes colocalize for the antibodies tested. Colocalized hemocytes can be seen in yellow. In groups exposed to pollutants, an increase in the number of immunoreactive hemocytes is observed. Strongly antibody-reactive epithelial cells are observed in the exposed groups (*). Use of the “display profile” function confirms the colocalization of antibodies.

**Figure 4 toxics-11-00731-f004:**
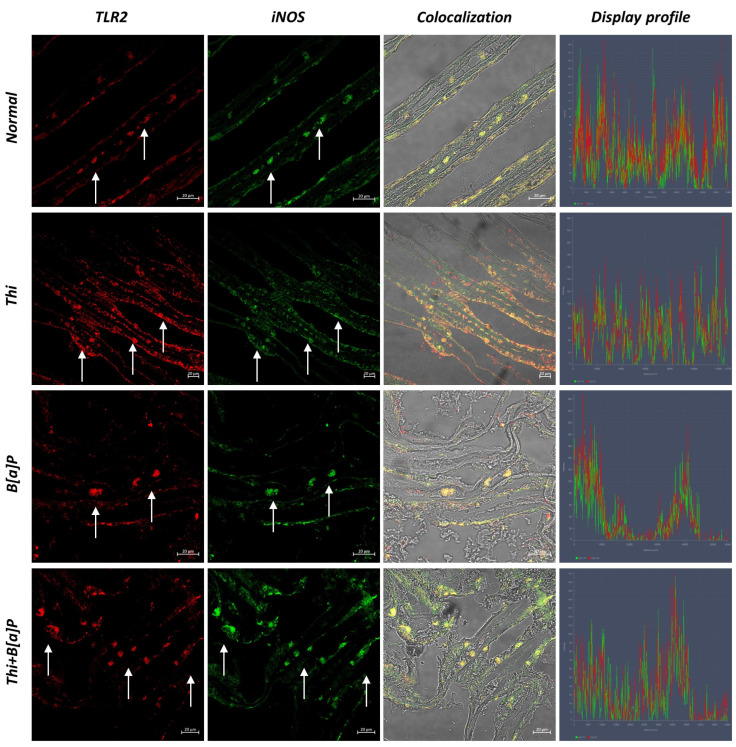
Cross section of gills of *M. galloprovincialis* (40×, scale bar: 20 µm) Immunofluorescence. Hemocytes with TLR2 (red) and iNOS (green) positivity were identified using confocal microscopy (arrow). For the tested antibodies, not all hemocytes colocalize. There is an increase in immunoreactive hemocytes among populations that have been exposed to pollutants. The “display profile” feature’s use validates the colocalization of antibodies.

**Figure 5 toxics-11-00731-f005:**
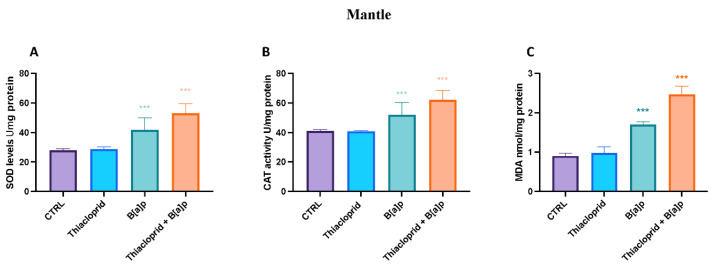
Antioxidant biomarkers (**A**,**B**) and lipid peroxidation (**C**) in the mantle of *M. galloprovincialis* exposed to thiacloprid, B[a]P and thiacloprid + B[a]P. The values are presented as the means ± SD; n = 10; The asterisk denotes a statistically significant difference when compared with the CTRL: *** *p*< 0.0001 versus control.

**Figure 6 toxics-11-00731-f006:**
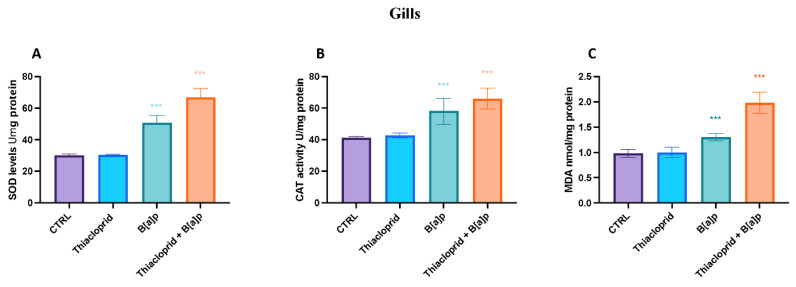
Antioxidant biomarkers (**A**,**B**) and lipid peroxidation (**C**) in gills of *M. galloprovincialis* exposed to thiacloprid, B[a]P and thiacloprid + B[a]P. The values are presented as the means ± SD; n = 10; The asterisk denotes a statistically significant difference when compared with the CTRL: *** *p*< 0.0001 versus control.

**Table 1 toxics-11-00731-t001:** Antibodies data.

Antibody	Supplier	Product Number	Dilution	Animal Source
TLR2	Active Motif, La Hulpe, Belgium	40981	1:125	rabbit
iNOS	Santa Cruz Biotechnology, Inc., Dallas, TX, USA	Sc-7271	1:200	mouse
Alexa Fluor 488 Donkey anti-Mouse IgG FITC conjugated	Molecular Probes, Invitrogen, Eugene, OR, USA	A21202	1:300	donkey
Alexa Fluor 594 Donkey anti-Rabbit IgG TRITC conjugated	Molecular Probes, Invitrogen, Eugene, OR, USA	A21207	1:300	donkey

**Table 2 toxics-11-00731-t002:** Statistical results (mean values ± standard deviations; *n* = 3).

	Normal	Thiacloprid	B[a]P	Thiacloprid + B[a]P
*Mantle*				
TLR2	345.72 ± 68.92 *	469.82 ± 64.21 **	471.69 ± 31.64 **	604.57 ± 63.07 *
iNOS	273.48 ± 65.38 *	436.45 ± 43.41 *	443.54 ± 53.87 **	592.35 ± 92.31 **
TLR2 + iNOS	212.30 ± 49.34 **	401.03 ± 37.24 *	416.24 ± 44.59 *	532.18 ± 83.12 **
*Gills*				
TLR2	312.31 ± 54.23 **	451.29 ± 61.94 *	458.46 ± 37.14 **	622.98 ± 63.42 *
iNOS	251.78 ± 35.29 **	423.37 ± 51.27 *	432.43 ± 20.83 *	598.23 ± 96.12 **
TLR2 + iNOS	201.02 ± 17.03 *	398.39 ± 47.91 **	406.90 ± 23.74 **	542.71 ± 45.82 *

** *p* ≤ 0.01, * *p* ≤ 0.05.

## Data Availability

Not applicable.

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
