# Peer review of "Internal Defense System of Mytilus galloprovincialis (Lamarck, 1819): Ecological Role of Hemocytes as Biomarkers for Thiacloprid and Benzo[a]Pyrene Pollution"

_toxics, 2023, doi:10.3390/toxics11090731_

Round 1
Reviewer 1 Report
In the Material and Methods section, it is necessary to add information on the content of the pollutants (thiacloprid and benzo[a]pyrene) in natural water and the studied tissues of the experimental mollusks.
It is necessary to explain the duration of the experiment (7 days) in terms of the immune response of the molluscs.
It is necessary to clarify why the mollusks were not fed during the experiment. The authors used natural water, is there enough food for molluscs in it?
In the Discussion section, the text contains a lot of repetition, similar information has already been presented in the Introduction section. It is necessary to shorten the text (about 1.5 pages) in the Discussion section and pay more attention to your own results. There is no discussion of the studied biomarkers. Interesting information has been obtained about the response of mollusks to the action of the pollutants (thiacloprid and benzo[a]pyrene), but, unfortunately, the authors do not discuss them.
Author Response
In the Material and Methods section, it is necessary to add information on the content of the pollutants (thiacloprid and benzo[a]pyrene) in natural water and the studied tissues of the experimental mollusks.
- We Thank the reviewer for the comment. We added these in the introduction section
It is necessary to explain the duration of the experiment (7 days) in terms of the immune response of the molluscs.
- We thank the reviewer for the comment. The purpose of this study was to observe the synergistic effect of two widely used environmental contaminants. Seven days of exposure were chosen as "acute exposure" as seen in previous studies (https://doi.org/10.1016/j.scitotenv.2022.153626, https://doi.org/10.1016/j.envpol.2023.121878)
It is necessary to clarify why the mollusks were not fed during the experiment. The authors used natural water, is there enough food for molluscs in it?
- We thank the reviewer for the comment. As we stated in lines 138-139: “Seawater and stock solutions were refreshed every 48h to guarantee a constant supply of fresh brackish food and prevent famine while exposed for an extended period”; the amount of nutrients in the water of this salt lake in Messina (Sicily) is more than enough to sustaine filter feeders.This acclimatation protocol is widley used as reported from previous studies conducted in the same area with the same animal model (https://doi.org/10.1016/j.scitotenv.2022.153626, https://doi.org/10.1016/j.envpol.2023.121878)
In the Discussion section, the text contains a lot of repetition, similar information has already been presented in the Introduction section. It is necessary to shorten the text (about 1.5 pages) in the Discussion section and pay more attention to your own results. There is no discussion of the studied biomarkers. Interesting information has been obtained about the response of mollusks to the action of the pollutants (thiacloprid and benzo[a]pyrene), but, unfortunately, the authors do not discuss them.
- We thank the reviewer for the comment.The text has been reorganized in both introduction and discussion. Some parts have been eliminated, others modified. The biomarkers studied are haemocytes and they are discussed in ll.364-425, where the defensive molecules expressed by haemocytes in relation to pollutants and the use of these cells as environmental biomarkers were analyzed.
Reviewer 2 Report
The paper reports on the effect of thiacloprid (a type of neonicotinoid insecticide) and benzo[a]pyrene (B[a]P) (a type of Polycyclic aromatic hydrocarbons, PAHs) on the hemocytes of the mussel Mytilus galloprovincialis. On the mantles and the gills of normal/control and exposed mussels (single and combined exposure to thiacloprid and BaP), the Authors performed:
- histological analysis by three different methods
- Immunohistochemical analysis using anti-TLR2 and anti-iNOS commercial antibodies
- oxidative stress biomarkers evaluation measuring SOD, CAT, and MDA activities
General comments
The manuscript is interesting but should be improved especially in the Results section.
In the Introduction as well as in the Discussion please follow the same order describing pesticide first and Polycyclic aromatic hydrocarbons (PAHs) second (or vice versa, your choice) but please follow the selected order in all the sections of the manuscript. For example, In the Introduction the sentences from Line 42 to Line 46 (Polycyclic aromatic hydrocarbons (PAHs) are …………………) could be moved before Line 64 (Benzo[a]pyrene (B[a]P) is an isomer belonging to the (PAHs) a………………..). Please check all over the manuscript.
Specific comments
- In 2. Materials and Methods section, did you feed the mussels? Please add this information.
- Line 146, The Authors wrote “Following the procedures for sample preparation for light microscopy, M. galloprovincialis samples were processed” please, add some information on the protocol used to include samples in paraffin and/or add a bibliographic reference.
-Line 198, The Authors wrote “mean ± standard error (SD)” please correct, is it standard error (SE) or standard deviation (SD)?
- In the section 3.1. Histological analysis, I suggest adding a brief description (maybe also some schemes) of the mantle and gill of a normal mussel to better guide non expert readers of the model used in this study.
- line 215-216, The Authors wrote “Infiltrates of haemocytes appear evident in the exposed groups” I suggest adding a symbol to indicate Infiltrated haemocytes in Figure 1.
-Legend of Figure 1, please add in the legend the measure of the bars that are present in the images of Figure 1. The “C” in the image on the line Normal/Masson stay for???? Please add information or delete.
-Legend of Figure 2, please add in the legend the measure of the bars that are present in the images of Figure 2.
- Figure 2, Is it possible to orient all the sections in the same way? I think this can help while observing and comparing the images shown in Figure 2.
- Section 3.2, Figure 3 and 4, i.e., mantle and gill immunohistochemical analysis, should be better described.
-Legend of Figure 3:
- The Authors wrote “Not all haemocytes colocalize for the antibodies tested” Is it possible to indicate some of these haemocytes in the images?
- please add in the legend the measure of the bars, it is indicated in the images 20 mm but it is too little for reading.
In Table 2, The Authors showed the numbers of immunoreactive haemocytes to TLR2 and iNOS. It could be interesting to add the row data of the counts as supplementary materials. In addition, did the Authors count the total haemocytes for each mussel analysed?
Line 293-295, For the sentence starting with Hydrocarbons, particularly polycyclic …………….. please add a reference).
English language seems to me fine
Author Response
The paper reports on the effect of thiacloprid (a type of neonicotinoid insecticide) and benzo[a]pyrene (B[a]P) (a type of Polycyclic aromatic hydrocarbons, PAHs) on the hemocytes of the mussel Mytilus galloprovincialis. On the mantles and the gills of normal/control and exposed mussels (single and combined exposure to thiacloprid and BaP), the Authors performed:
- histological analysis by three different methods
- Immunohistochemical analysis using anti-TLR2 and anti-iNOS commercial antibodies
- oxidative stress biomarkers evaluation measuring SOD, CAT, and MDA activities
General comments
The manuscript is interesting but should be improved especially in the Results section.
In the Introduction as well as in the Discussion please follow the same order describing pesticide first and Polycyclic aromatic hydrocarbons (PAHs) second (or vice versa, your choice) but please follow the selected order in all the sections of the manuscript. For example, In the Introduction the sentences from Line 42 to Line 46 (Polycyclic aromatic hydrocarbons (PAHs) are …………………) could be moved before Line 64 (Benzo[a]pyrene (B[a]P) is an isomer belonging to the (PAHs) a………………..). Please check all over the manuscript.
- We thank the reviewer for the comment. Text has been reorganized, as suggested.
Specific comments
- In 2. Materials and Methods section, did you feed the mussels? Please add this information.
- We Thank the reviewer for the comment. As we stated in lines 138-139: “Seawater and stock solutions were refreshed every 48h to guarantee a constant supply of fresh brackish food and prevent famine while exposed for an extended period”; the amount of nutrients in the water of this salt lake in Messina (Sicily) is more than enough to sustaine filter feeders.This acclimatation protocol is widley used as reported from previous studies conducted in the same area with the same animal model (https://doi.org/10.1016/j.scitotenv.2022.153626, https://doi.org/10.1016/j.envpol.2023.121878)
- Line 146, The Authors wrote “Following the procedures for sample preparation for light microscopy, M. galloprovincialis samples were processed” please, add some information on the protocol used to include samples in paraffin and/or add a bibliographic reference.
- We thank the reviewer for the comment. A sentence and related bibliographic reference have been added.
-Line 198, The Authors wrote “mean ± standard error (SD)” please correct, is it standard error (SE) or standard deviation (SD)?
- We thank the reviewer for the comment. We fixed the typing error
- In the section 3.1. Histological analysis, I suggest adding a brief description (maybe also some schemes) of the mantle and gill of a normal mussel to better guide non expert readers of the model used in this study.
- We thank the reviewer for the comment. Further information about normal mantle and gills histology has been added in the discussion section.
- line 215-216, The Authors wrote “Infiltrates of haemocytes appear evident in the exposed groups” I suggest adding a symbol to indicate Infiltrated haemocytes in Figure 1.
- We thank the reviewer for the comment. We fixed it as suggested
-Legend of Figure 1, please add in the legend the measure of the bars that are present in the images of Figure 1. The “C” in the image on the line Normal/Masson stay for???? Please add information or delete.
- We thank the reviewer for the comment.Done
-Legend of Figure 2, please add in the legend the measure of the bars that are present in the images of Figure 2.
- We thank the reviewer for the comment.Done
- Figure 2, Is it possible to orient all the sections in the same way? I think this can help while observing and comparing the images shown in Figure 2.
- We thank the reviewer for the comment. Unfortunately, it is not possible to change the orientation of the photos because the confocal microscope image capture program does not allow you to change the orientation of the images without impacting the scale bar, which is automatically entered immediately after the shot is taken.
- Section 3.2, Figure 3 and 4, i.e., mantle and gill immunohistochemical analysis, should be better described.
- We thank the reviewer for the comment. Our description focuses on haemocytes, which are the object of our immunohistochemical investigation. Nevertheless, epithelial cells were also positive for the antibodies tested, as already enunciated.
-Legend of Figure 3:
- The Authors wrote “Not all haemocytes colocalize for the antibodies tested” Is it possible to indicate some of these haemocytes in the images?
- We thank the reviewer for the comment. Colocalized haemocytes can be seen in yellow. This sentence has been added in the figure 3 legend.
- please add in the legend the measure of the bars, it is indicated in the images 20 mm but it is too little for reading.
- We thank the reviewer for the comment.We added it as suggested
In Table 2, The Authors showed the numbers of immunoreactive haemocytes to TLR2 and iNOS. It could be interesting to add the row data of the counts as supplementary materials. In addition, did the Authors count the total haemocytes for each mussel analysed?
- We thank the reviewer for the comment.Ten sections and twenty fields per sample were examined to collect data for the quantitative analysis, both for mantle and gills for each group. We believe that adding all the raw data (very numerous) may create further confusion and is not strictly necessary, but we greatly appreciate the suggestion and will try to implement it in our next study.
Line 293-295, For the sentence starting with Hydrocarbons, particularly polycyclic …………….. please add a reference).
- We thank the reviewer for the comment. This sentence has been removed.
Reviewer 3 Report
The paper entitled "Internal Defense System of Mytilus galloprovincialis (Lamarck, 1819): Ecological Role of Haemocytes as Biomarkers for Thiacloprid and
Benzo[a]Pyrene Pollution" is a well written manuscript of a project with high significance of content. The study is well designed and methods are adequately described. The way of presentation makes everything clear to the reader even those not familiar with the field.
I have only some minor corrections to suggest
1. The experience of authors in the field of ecotoxicological studies is obvious. The self citation is not necessary since the reference list is already long. In my opinion some of the papers of this team should be eliminated.
2. There are two points for editing in the text: Line 79, the concentration of the PAHs is g/g? Also in line 200, post hoc should be corrected.
3. TLR and iNOS abbreviations must be explained in the abstract.
Author Response
The paper entitled "Internal Defense System of Mytilus galloprovincialis (Lamarck, 1819): Ecological Role of Haemocytes as Biomarkers for Thiacloprid and
Benzo[a]Pyrene Pollution" is a well written manuscript of a project with high significance of content. The study is well designed and methods are adequately described. The way of presentation makes everything clear to the reader even those not familiar with the field.
I have only some minor corrections to suggest
- The experience of authors in the field of ecotoxicological studies is obvious. The self citation is not necessary since the reference list is already long. In my opinion some of the papers of this team should be eliminated.
- Thank you for the kind words. However, we believe that in this context the quotations included are useful to corroborate the data obtained from our study.
- There are two points for editing in the text: Line 79, the concentration of the PAHs is g/g? Also in line 200, post hoc should be corrected.
- We thank the reviewer for the comment. We fixed the text as suggested.
- TLR and iNOS abbreviations must be explained in the abstract.
- We thank the reviewer for the comment. We fixed the text as suggested.
Round 2
Reviewer 1 Report
-